# Unusual metastases from differentiated thyroid cancers: A multicenter study in Korea

Jee Hee Yoon[ID]<sup>1</sup>, Min Ji Jeon<sup>2</sup>, Mijin Kim<sup>3</sup>, A. Ram Hong<sup>1</sup>, Hee Kyung Kim<sup>1</sup>, Dong Yeob Shin<sup>4</sup>, Bo Hyun Kim<sup>3</sup>, Won Bae Kim<sup>2</sup>, Young Kee Shong<sup>2</sup>, Ho-Cheol Kang[ID]<sup>1</sup>*

**Jee Hee Yoon**[1], **Min Ji Jeon**[2], **Mijin Kim**[3], **A. Ram Hong**[1], **Hee Kyung Kim**[1], **Dong Yeob Shin**[4], **Bo Hyun Kim**[3], **Won Bae Kim**[2], **Young Kee Shong**[2], **Ho-Cheol Kang**[1]*

**1** Division of Endocrinology and Metabolism, Department of Internal Medicine, Chonnam National University Medical School, Gwangju, South Korea, **2** Division of Endocrinology and Metabolism, Department of Internal Medicine, Asan Medical Center, University of Ulsan College of Medicine, Seoul, South Korea, **3** Division of Endocrinology and Metabolism, Department of Internal Medicine, Biomedical Research Institute, Pusan National University Hospital, Pusan, South Korea, **4** Division of Endocrinology and Metabolism, Department of Internal Medicine, Severance Hospital, Yonsei University College of Medicine, Seoul, South Korea

* drkang@chonnam.ac.kr

## Abstract

### Background

Although infrequent, distant metastasis from differentiated thyroid cancer is the main cause of mortality in patients and mostly involves the lung, bone, and brain. Distant metastases to other sites in differentiated thyroid cancer patients are rare, thus, the clinical course of unusual metastases has not been adequately researched. In the present study, the clinico-pathological findings and treatment outcomes of unusual metastases in differentiated thyroid cancer patients in Korea were evaluated.

### Patients and methods

We retrospectively reviewed the medical records of differentiated thyroid cancer patients with unusual metastases in four Korean tertiary hospitals (Chonnam National University Hwasun Hospital, Asan Medical Center, Busan National University Hospital, Severance Hospital). Unusual metastases were diagnosed using (1) cytology or histology and/or (2) imaging studies including fluorodeoxyglucose F 18 positron emission tomography/computed tomography and/or iodine 131 whole body scans with simultaneously elevated serum levels of thyroglobulin. The pathological findings of primary thyroid cancer, diagnostic method for unusual metastases, and treatment responses of unusual metastases were examined.

### Results

In all, 25 unusual metastatic foci of 19 patients were analyzed; 13 patients (68.4%) had papillary thyroid carcinoma including 4 follicular variant papillary thyroid carcinomas. The median time interval between the first diagnosis of primary thyroid cancer and unusual metastases diagnosis was 110 months (11.0–138.0 months). Only 4 patients (21.1%) had synchronous unusual metastases and 6 patients (31.6%) were symptomatic. Unusual metastases included 19 metastases to solid organs (6 to kidney, 5 to liver, 4 to pancreas, 3

**Data Availability Statement:** All relevant data are within the manuscript.

**Funding:** The author(s) received no specific funding for this work.

**Competing interests:** The authors have declared that no competing interests exist.

to adrenal gland, and 1 to ovary) and 6 to the skin and muscles. Unusual metastases were pathologically proven in 10 patients (52.6%) and 11 of 16 patients (68.8%) who received iodine 131 whole body scans had radioiodine-refractory differentiated thyroid cancer. Among 5 patients treated with tyrosine kinase inhibitors, 4 treated with lenvatinib showed stable disease or a partial response at the first treatment response. Six patients (31.6%) died due to disease progression during the median 20.0-month follow-up period (11.0–55.0 months).

## Conclusion

Unusual metastases from differentiated thyroid cancer are thought to be underestimated due to disease rarity and their metachronous nature with other distant metastases. The most of unusual metastases in differentiated thyroid cancer patients are existed with usual distant metastasis and clinical outcomes of those could not be significantly different from the prognosis of usual distant metastasis.

## Introduction

Differentiated thyroid cancers (DTCs) have indolent clinical course and good prognosis with an approximate 85–90% 10-year survival rate [1–3]. Distant metastases from DTCs are uncommon but are one of the main causes of cancer-specific mortality in DTC patients [4]. The 10-year-survival rate is significantly decreased to 10% due to loss of radioiodine avidity of distant metastasis [3, 5]. Distant metastases of DTC are simultaneously observed in approximately 3–5% of patients at the first diagnosis of DTC, which increases up to 20% when a metachronous occurrence pattern is observed [6–8]. Therefore, early detection and appropriate management of distant metastases are critical for better clinical outcomes in patients with advance thyroid cancer.

Distant metastases from DTC mainly involve the lung, bone, and brain [7]. Metastases to other sites from DTC are extremely rare; therefore, the majority of unusual metastases (UMs) have been published as case reports and small case series [9]. The reported incidence of UM in DTC patients is <1% [10], however, UMs might easily be overlooked, particularly when asymptomatic. Recent progress in imaging can increase the detection rate of UM in DTC patients. Due to diverse metastatic sites and disease rarity, previously reported data on the prognosis of UMs are limited and inconsistent [11, 12]. In the present study, we analyzed the clinical characteristics of UMs and evaluated the proper diagnostic and management methods based on metastatic sites and patient status in patients with UMs from DTC.

## Patients and methods

### Patients

We screened DTC patients who had been treated between January 2000 and August 2016 from four tertiary hospitals in Korea (Chonnam National University Hwasun Hospital, Asan Medical Center, Busan National University Hospital, Severance Hospital) and retrospectively reviewed the medical records of DTC patients with distant metastasis. Distant metastases were divided to two groups; usual metastasis and UM. A UM was defined as a distant metastasis to sites excluding the lung, bone, and brain. Skin and muscle metastases were included only if they were not adjacent to the primary thyroid cancer. Diagnoses were made using one of the

following methods: pathology based on fine-needle aspiration cytology (FNAC) and biopsy or imaging using [18]fluoro-deoxy-glucose positron emission tomography/computed tomography ([18]FDG PET/CT) scans or [131]iodine whole body scans ([131]I WBS) with simultaneously elevated serum levels of thyroglobulin (Tg). Synchronous UM was defined when the time interval from the first diagnosis of primary thyroid cancer to UM diagnosis was <6 months. Radioactive iodine (RAI) refractoriness was defined as a lack of RAI uptake or disease progression after RAI therapy within the previous 12 months or cumulative RAI dose $\geq$ 600 mCi [13]. When anti-Tg antibody was negative according to each institution's reference value, serum levels of Tg were included in analyses.

All patients underwent total thyroidectomy, and pathological findings of primary thyroid cancer according to AJCC/IUCC (8th edition) [14] and clinical characteristics were recorded. Pathological diagnoses of UMs were obtained using FNAC and biopsy with ultrasonography (US) or computed tomography (CT) guidance, or surgery. Therapeutic modalities included RAI ablation therapy for RAI avid lesions, radiofrequency ablation (RFA), surgery, systemic chemotherapy, and tyrosine kinase inhibitors (TKIs). The treatment efficacy of TKI therapy was divided into four categories: complete response (CR), partial response (PR), stable disease (SD), and progressive disease (PD) according to the RECIST version 1.1 criteria [15]. Safety and adverse events (AEs) were monitored according to each institution's guideline. The TKI dose was reduced when drug-related AEs were intolerable. This study was approved by the institutional review boards of the participating hospitals (IRB numbers: CNUHH 2020–131, 4-2020-0626, 2005-001-089, and 2019–1453). All data were fully anonymized before assessment and IRB waived the requirement for informed consent.

### Statistical analyses

Data are expressed as median (interquartile range) or n (%), and continuous variables were analyzed using Student's *t*-test. Overall survival (OS) was evaluated using Kaplan-Meier estimates with 95% confidence intervals (CIs). All statistical analyses were performed using SPSS Statistics, (IBM, Armonk, NY, USA) and a *p*-value < 0.05 was considered statistically significant.

## Results

### Baseline characteristics of UMs in DTC patients

Among investigated total of 38,772 DTC patients, 442 usual distant metastases (1.1%) in 392 patients (321 cases of lung metastases, 105 cases of bone metastases, and 16 cases of brain metastases) and 25 UMs (0.06%) in 19 patients were observed. The median age of the 19 DTC patients with UMs was 68.0 years and most cases were male (68.4%). The most frequent histological type of primary thyroid cancer was PTC with 4 cases of the follicular variant type. Among 16 patients with available TMN stage data of primary thyroid cancer, 9 (43.8%) had distant metastasis at the time of primary thyroid cancer surgery. Only one papillary thyroid microcarcinoma was included and distant metastasis was already observed at the first diagnosis of primary thyroid cancer (Table 1).

In the 19 patients, 25 unusual metastatic foci were observed and 19 UMs (76.0%) involved solid organs. Among the solid organ UMs (n = 19), metastasis to the kidney (n = 6) was the most frequent, followed by the liver (n = 5) and the pancreas (n = 4). The muscle metastases (n = 4) involved the vastus lateralis (n = 2), glutes medius (n = 1), and infraspinatus muscles (n = 1). Skin metastases (n = 2) were in the chest wall. Only 6 of the 19 patients (31.6%) had definitive symptoms associated with UMs and 4 patients (21.1%) had synchronous UMs. At the time of UM diagnosis, all but one patient (94.7%), who had only muscle metastasis, had

**Table 1. Baseline characteristics of DTC patients with UM (n = 19).**

| | N,% |
|---|---|
| Median age at time of UM diagnosis (years) (IQR) | 68.0 (59.0–76.0) |
| Sex, male (%) | 13 (68.4) |
| Histology of primary thyroid cancer | |
| PTC | 13 (68.4) |
| Classic | 9 (47.4) |
| Follicular variant | 3 (15.8) |
| Classic and follicular variant | 1 (5.3) |
| FTC | 6 (31.6) |
| TMN stage (8th edition) (n = 16)* | |
| *T stage* | |
| T1a | 1 (5.3) |
| T1b | 0 (0.0) |
| | 6 (31.6) |
| T3 | 7 (36.8) |
| T4 | 2 (10.5) |
| *N stage* | |
| Nx | 4 (25.0) |
| N0 | 2 (12.5) |
| N1a | 3 (18.8) |
| N1b | 7 (43.8) |
| *M stage* | |
| M0 | 7 (56.3) |
| M1 | 9 (43.8) |

DTC: differentiated thyroid cancer; UM: unusual metastasis; PTC: papillary thyroid cancer; FTC: follicular thyroid cancer; RAI: radioactive iodine; IQR: interquartile range

*The 16 patients had TNM staging results of primary thyroid cancer

distant metastases to usual metastatic sites. Simultaneous lung metastases were most frequent and 3 patients (15.8%) had multiple metastases to the lung, bone, and brain. Among 16 patients who received diagnostic [131]I WBS, 11 (68.8%) showed RAI refractoriness. Among 12 patients who were diagnosed based on [18]FDG PET/CT scan, 8 (66.7%) showed FDG avidity. Only one patient with kidney and muscle metastases showed simultaneous RAI and FDG uptake in the metastatic lesion in the kidney (Table 2).

## Pathological diagnostic approach for UM

UMs of 10 patients (52.6%) were pathologically diagnosed based on cytological or histological results. One patient with pancreatic head metastasis was diagnosed using endoscopic US (EUS)-guided FNA and the other patient with pancreatic body and tail metastases was diagnosed using CT-guided FNA. The histological diagnosis was performed using biopsy (n = 5) including US-guided biopsy and excision biopsy. Diagnostic surgery was performed for ovary metastasis (n = 1) (Table 3).

## Treatment strategies for UM

Among the 19 patients with UM, 4 (21.1%) received only supportive care without any intervention due to poor patient condition or economic problems. UM treatments included local

**Table 2. Clinical characteristics of UM in DTC (n = 19).**

|  | N, % |
|---|---|
| Symptomatic | 6 (31.6) |
| Synchronous | 4 (21.1) |
| Time interval between the first diagnosis of DTC and UM diagnosis (months) (IQR) | 110 (11.0–138.0) |
| Largest UM diameter at diagnosis (cm), (IQR) | 3.0 (2.0–5.9) |
| Location (25 metastatic foci)* |  |
| Solid organ | 19 (76.0) |
| Kidney | 6 (24.0) |
| Liver | 5 (20.0) |
| Pancreas | 4 (16.0) |
| Adrenal gland | 3 (12.0) |
| Ovary | 1 (4.0) |
| Muscle | 4 (16.0) |
| Skin | 2 (8.0) |
| Distant metastasis at the time of UM diagnosis | 18 (94.7) |
| Lung only | 9 (47.4) |
| Bone only | 1 (5.3) |
| Lung and bone | 4 (21.1) |
| Lung and brain | 1 (5.3) |
| Lung, bone, and brain | 3 (15.8) |
| RAI refractoriness at the time of UM diagnosis (n = 16)** | 11 (68.8) |
| Lack of RAI uptake | 4 (25.0) |
| Progression despite RAI therapy in 12 months or total RAI dose > 600 mCi | 4 (25.0) |
| Both | 3 (18.8) |
| FDG avidity of UM (n = 12)*** | 8 (66.7) |

UM: unusual metastasis; RAI: radioactive iodine; MRI: magnetic resonance imaging; CT: computed tomography; FDG: fluoro-deoxy-glucose; IQR: interquartile range; FNA: fine-needle aspiration; RFA: radiofrequency ablation; TKI: tyrosine kinase inhibitor

*The 25 metastatic foci of 19 patients were observed. Five patients had more than two UMs.

**The 16 patients diagnosed using $I^{131}$ whole body scan (WBS) were analyzed for RAI refractoriness

***The 12 patients diagnosed with UMs based on $^{18}$FDG PET/CT scans

treatment for isolated metastasis and systemic treatment. Two patients (one with skin metastasis on the chest wall, and one with thigh muscle metastasis) had surgeries and both metastatic lesions did not recur during the follow-up period. The patient who underwent diagnostic oophorectomy for ambiguous ovarian lesion had no recurrence in the ovary. Two patients with renal metastases underwent RFA and both lesions shrunk and the disease remained stable. Among 10 patients who had systemic treatment, 4 underwent RAI therapy. When therapeutic $^{131}$I WBS was performed for lung metastases, one male patient was found to have unsuspected thigh muscle metastasis and the US-guided FNA of the muscle lesion confirmed the metastasis was from PTC (Fig 1). CR of muscle metastasis was attained based on surgical specimens of post-RAI surgery for the remnant lesion. Five patients received TKI therapy for RR DTC. One patient with adrenal, renal, and hepatic metastases treated daily with 600 mg sorafenib had to discontinue TKI therapy due to grade 3 plantar-palmar erythrodysaesthesia (hand-foot syndrome) before the evaluation of drug response. Four patients were treated with lenvatinib. One PTC patient with lung metastasis was treated with lenvatinib therapy for newly detected hepatic metastasis as a sorafenib salvage therapy and the first treatment

**Table 3. Clinical course of UM in DTC patients (n = 19).**

| | |
|---|---|
| Diagnostic method for UM | |
| Pathology | 10 (52.6) |
| FNA | 4 (21.1) |
| Biopsy | 5 (31.3) |
| Surgical specimen | 1 (6.3) |
| Imaging studies | 9 (47.4) |
| RAI uptake ± CT/MRI | 2 (10.5) |
| 18-FDG uptake ± CT/MRI | 7 (36.8) |
| First-line therapy for UM | |
| Observation | 4 (21.1) |
| Local treatment | 5 (31.3) |
| Surgery (including excision) | 3 (15.8) |
| RFA | 2 (10.5) |
| Systemic treatment | 10 (52.6) |
| TKI therapy | 5 (31.3) |
| RAI ablation therapy | 4 (21.1) |
| Systemic chemotherapy | 1 (5.3) |
| Mortality rate | 6 (31.6) |
| Cancer-related mortality | 6 (31.6) |
| Treatment-related mortality | 0 (0.0) |
| Median follow-up months from UM diagnosis (IQR) | 20.0 (11.0–55.0) |

UM: unusual metastasis; RAI: radioactive iodine; FDG: fluoro-deoxy-glucose; RFA: radiofrequency ablation; TKI: tyrosine kinase inhibitor; IQR: interquartile range

response showed SD, however, PD was observed at 10 months after starting lenvatinib. Among three patients treated with lenvatinib therapy for pancreatic metastasis, two PTC patients showed marked tumor regression (Fig 2) and one FTC patient with lung and pleura metastases died due to disease progression although pancreatic metastasis showed SD. Table 4 summarizes the clinical characteristics and courses of TKI-naïve RR DTC patients with pancreatic metastasis treated with lenvatinib.

## Prognosis of UM

The cancer-related mortality was 31.6% (n = 6) in the 19 patients. Median OS was 68.0 months and mean OS was 58.3 months during the median 20.0-month follow-up period (11.0–55.0 months) (Fig 3).

## Discussion

UMs in DTC patients have been reported in only a few case reports and small series due to disease rarity, thus, clinical characteristics and disease course of UM have not been adequately researched [9, 12]. In a recent study, a low incidence of UM (36 of 3,982 DTC patients, 0.9%) and poor prognosis were reported, however, 12 cases of brain metastases were included, which could explain the high mortality [10]. The National Epidemiologic Survey of Thyroid cancer (NEST) study based on the Korea Central Cancer Registry demonstrated 34 distant metastases (0.6%) among total 5,796 thyroid cancer patients during 10.6 years of follow-up [16]. So, our data is in accordance with Korean nationwide data. The low incidence of distant metastases might be due to early surgery and high incidence of small papillary microcarcinoma in Korea.

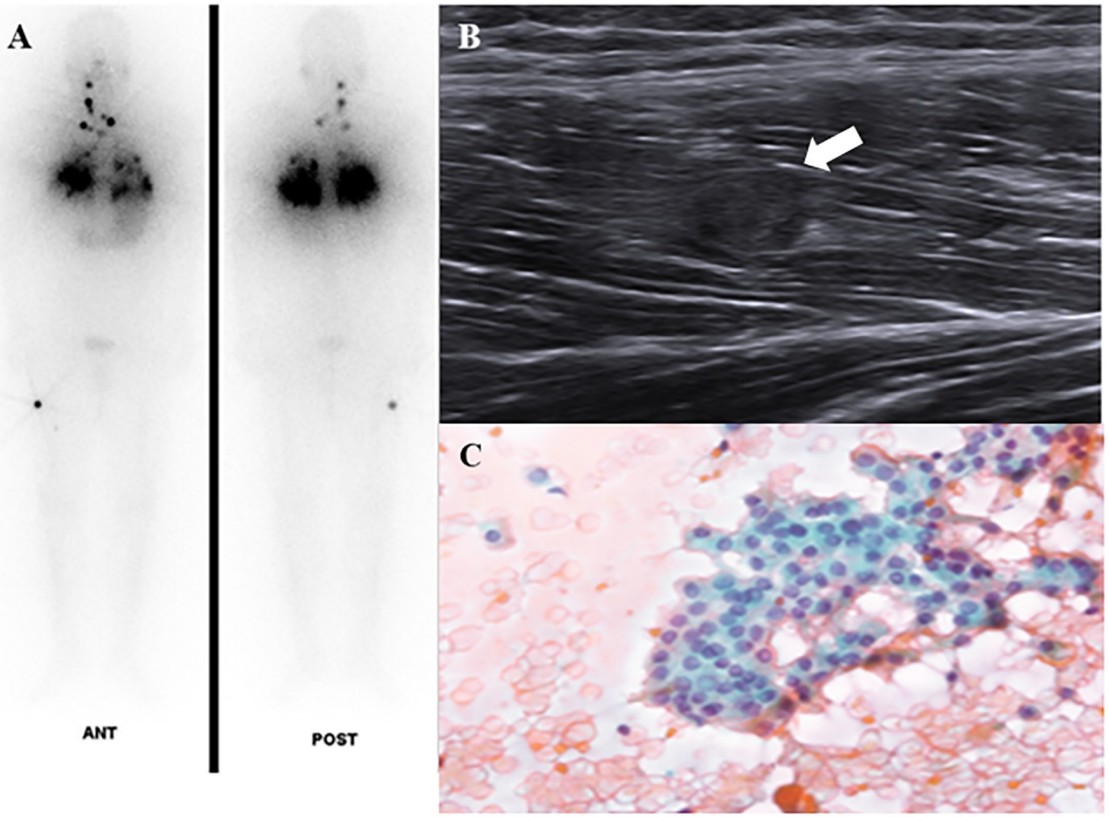

**Fig 1. A 34-year-old male patient with follicular variant-papillary thyroid cancer (FV-PTC) showed diffuse lung metastases and unusual metastases to the vastus lateralis muscle in the right thigh.** (A) [131]Iodine whole body scan showed radioactive iodine avidity in the anterior neck and metastatic lesions in the neck, both lungs, and right thigh muscles. (B) Ultrasonography showed a round hypoechoic lesion in the muscle of the right thigh. (C) Fine-needle aspiration indicated muscle metastasis from FV-PTC.

Some case reports and series have shown that synchronous UM in DTC patients presenting initially with UM-related symptoms, but they have difficulty in finding the primary cancer origin due to unusual metastatic sites [17]. However, the most of UMs from DTC were found to develop metachronously in previous studies [10, 12]. Only 4 patients (21.1%) in our study had synchronous UM, indicating a higher incidence of UMs in advanced thyroid cancer during long-term follow-up with diverse imaging modalities such as PET-CT, [131]I WBS, and CT scans due to a metachronous pattern. In a multicenter study in Korea, 5 of 57 (8.8%) TKI-naive RR DTC patients had UMs to the liver (n = 2), adrenal gland (n = 2), and kidney (n = 1) [18]. In previous studies, the focus has been on the metastatic site and OS, and UM treatment modalities have been limited to radiation therapy and mainly surgery for local control when the metastatic lesion showed RAI non-avidity [19–21]. When the long-term outcome of metastasectomy in thyroid cancer patients with liver and pancreatic metastasis was investigated, the effectiveness of surgery was demonstrated only for isolated hepatic metastasis [11]. However, UMs from DTC usually occur in advanced and aggressive thyroid cancer and other distant metastases exist simultaneously [22]. Surgery may be the best choice for patients with good general condition and isolated resectable UMs; however, minimally invasive procedures including RFA, microwave ablation and TKI therapy could be better treatment options for patients with co-morbidities or multiple metastases [23–25]. In the present study, 3 DTC patients with pancreatic metastasis treated with lenvatinib showed PR (n = 2) and SD (n = 1)

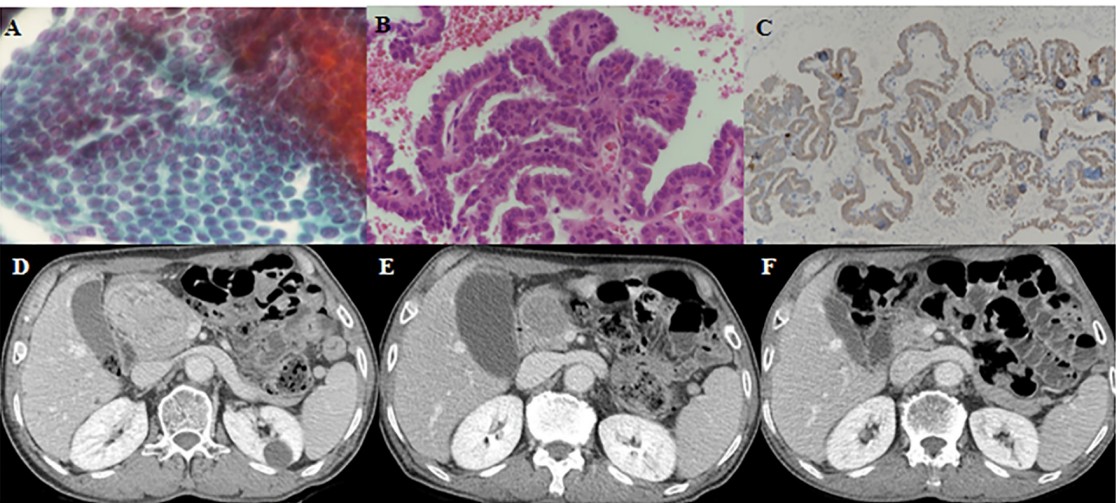

**Fig 2. A 71-year-old male patient presenting with pancreatic metastasis from papillary thyroid cancer.** (A, B, C) Endoscopic ultrasonography-guided fine-needle aspiration (FNA) and biopsy showed metastatic PTC on the pancreatic head mass. (A): Hematoxylin and eosin (H-E) staining of FNA (x 400), (B): H-E staining of biopsy (x400), (C): BRAFV600E staining, (D) Abdomen computed tomography (CT) showed 6.9 cm heterogeneously enhancing mass on the pancreatic head before lenvatinib therapy. (E) Follow-up abdomen CT at 3 months after lenvatinib therapy showed the pancreatic tumor decreased in size from 6.9 cm to 4.1 cm. (F) Follow-up abdomen CT at 24 months after lenvatinib therapy showed that the pancreatic tumor was significantly reduced to 1.2 cm.

at the time of the best response. The UMs involved a broad spectrum of metastatic sites; thus, diagnostic and treatment strategies based on metastatic site are necessary.

We diagnosed UMs via cytology and/or histology and imaging using $^{18}$FDG PET/CT scan or $^{131}$I WBS. FNA or biopsy for newly detected UMs are not mandatory in patients with already widespread multiple metastases, however, pathological diagnosis is helpful to exclude other diseases in some cases. If procedure-related risks and expenses are acceptable,

**Table 4. Treatment response of three TKI-naive patients with pancreatic metastasis treated with lenvatinib.**

|  | Case 1 | Case 2 | Case 3 |
|---|---|---|---|
| Age/Sex | 74/M | 61/M | 68/F |
| Initial TNM stage (8th edition) | T3NxM0 | T3N1bM0 | T3N1bM1 |
| Symptom | Yes | No | Yes |
| Other metastasis | Lung | Lung, bone | Lung |
| FDG avidity | Yes | NA | No |
| Largest UM diameter at diagnosis (cm) | 6.0 | 7.0 | 5.4 |
| Largest UM diameter at best response (cm) | 1.5 | 2.5 | 5.1 |
| Serum levels of Tg before lenvatinib | 186.0 | 1615.0 | 249.7 |
| Serum levels of Tg at best response | 22.64 | 48.90 | 208.0 |
| Duration of lenvatinib therapy (months) | 26 | 23 | 5 |
| Treatment response | PR = >SD | PR = >SD | SD = >PD |
| Treatment-related AEs | Proteinuria, pancreatic enzyme elevation | Proteinuria, HTN, blood glucose elevation | HTN, edema, constipation |
| Follow-up duration from UM diagnosis (months) | 36 | 23 | 16 |
| Mortality | Alive | Alive | Dead |

FDG: fluoro-deoxy-glucose; NA: not assessed; UM: unusual metastasis; Tg: thyroglobulin: PR: partial response; SD: stable disease; PD: progressive disease; AEs: adverse events; HTN

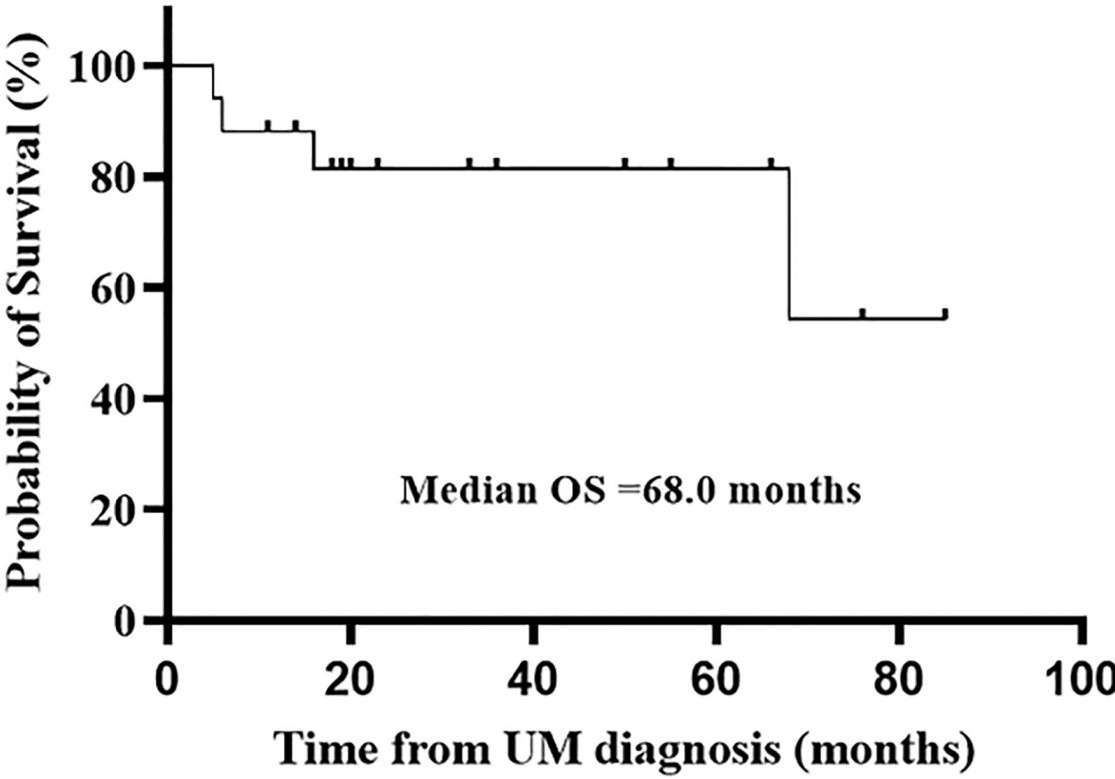

**Fig 3. Overall survival (OS) in patients with unusual metastases based on Kaplan-Meier curve.**

pathological diagnosis can be considered when treatment modalities can be modified with accurate UM diagnosis. In the present study, 52.6% of the 19 patients were pathologically diagnosed mainly using US-guided FNA or biopsy. In patients with pancreatic metastasis, a diagnostic method was selected based on pancreatic tumor location. EUS-guided FNA was used for UMs to the pancreatic head and CT-guided FNA for UMs to the pancreas body and tail. Surgical or CT-guided biopsy has been used to diagnosis pancreatic metastasis in the past, however, EUS-guided FNA has been reported in recent studies as a potentially reliable diagnostic tool for pancreatic metastasis [26, 27]. UMs were detected in [18]FDG PET/CT scans or [131]I WBS and then additional imaging modalities including CT and MRI were performed to determine tumor size, invasiveness, and structural characteristics.

The treatment strategies for UMs are divided into systemic and local therapy based on patient condition, UM location, and thyroid cancer status. As a local therapy for isolated or locally controllable UMs, surgery (metastasectomy or excision) has been the treatment of choice and external beam radiation and RFA have been used in previous studies [11, 19, 28, 29]. In the present study, two excisions (one for a muscle metastasis and one for a skin metastasis on the chest wall), one diagnostic oophorectomy, and two RFAs for renal metastases were performed as local treatment. Targeted tumor regression and stabilization were observed in all patients treated using local therapy. Systemic therapy is considered when a patient has widespread disease or involvement of an inoperable critical organ. The mainstream systemic therapy for metastasis in DTC patients is RAI therapy, however, before the development of TKIs, treatment options were not available when metastatic lesions showed RAI non-avidity [5]. Several systemic chemotherapies have been attempted, which are not recommended for RR DTC due to serious AEs with no survival benefit [30]. In the present study, one patient who

underwent systemic chemotherapy for renal metastasis discontinued treatment due to disease progression. Since approval for TKI therapy for advanced thyroid cancer, the use of TKIs in advanced thyroid cancer patients with UM has been reported in several cases [11]. Previously reported cases treated with sorafenib have shown shortly delayed disease progression but unclear survival benefit in patients with liver and pancreatic metastases from advanced thyroid cancer [31, 32]. However, in a recent case report of a PTC patient with multiple simultaneous metastases to the pancreas, kidney, gluteus maximus muscle, and lung who was treated with lenvatinib, SD was observed during follow-up 13 months after starting lenvatinib [33]. In our series, five patients received TKI therapy and four patients with lenvatinib showed a treatment response. Among three TKI-naïve patients with pancreatic metastases were treated with lenvatinib, one patient died due to disease progression of other metastatic lesions although drug response of pancreas metastasis showed SD. The other two patients showed PR at the first response evaluation and maintained SD. During lenvatinib therapy, well-known AEs including hypertension, proteinuria, and fatigue were observed. In addition, rare AEs associated with pancreatic tumor treatment including elevated pancreatic enzymes and blood glucose were observed in patients with pancreatic metastases, which were manageable with supportive care and insulin therapy. When TKI therapy is initiated for UMs in DTC patients, rare drug-related AEs should be considered.

The prognosis of thyroid cancer patients with UM has not been adequately evaluated, and the data reported have been inconsistent [34, 35]. In a systematic review of rare metastases in DTC patients, the mean OS of 94 patients from 77 studies was 60 months [12], similar to the mean OS observed in the present study (58.3 months). In a recent study, a high cancer-specific mortality of 58.3% was observed (21 of 36 patients) during the median follow-up period of 13 months after UM diagnosis, however, the study included 12 patients with UM to the brain, of whom 9 died [10]. The survival data of UM in DTC patients are discordant based on UM location and treatment modality. In a systematic review that included 16 patients in 15 studies, the mean OS was 37.6 months in DTC patients with pancreatic or hepatic metastases and who received surgical resection [11]. Predicting prognosis in patients with UM is difficult. In addition, the prognosis of UM is associated with disease status of primary thyroid cancer such as simultaneous distant metastases to other organs and RAI refractoriness. In the present study, almost all patients, except one patient with isolated UM to the infraspinatus muscle, showed other distant metastases. Six patients (31.6%) in our study died due to disease progression and all had solid organ UMs. Three patients with multiple distant metastases including two brain metastases with synchronous UMs died 6 months after the first diagnosis of advanced thyroid cancer.

The present study had several limitations due to the retrospective study design. Primary thyroid cancer TNM staging and pathologic diagnosis of UMs were not obtained in all cohort patients and BRAF mutation analyses on pathologic specimens were performed in only a few patients. Regular follow-up measurements of serum levels of Tg and imaging studies were not performed with the same time interval due to multi-center involvement.

Despite the limitations, our results demonstrate the clinical course of UMs in DTC patients including diagnostic methods and several management modalities. Early detection and TKI therapy for selected DTC patients with UM could improve clinical outcomes.

## Author Contributions

**Conceptualization:** Hee Kyung Kim, Won Bae Kim, Young Kee Shong.

**Data curation:** Jee Hee Yoon, Min Ji Jeon, Mijin Kim, Dong Yeob Shin.

**Formal analysis:** Jee Hee Yoon, A. Ram Hong.

**Investigation:** Min Ji Jeon, Mijin Kim, Dong Yeob Shin.

**Project administration:** Ho-Cheol Kang.

**Resources:** Hee Kyung Kim.

**Supervision:** Hee Kyung Kim, Bo Hyun Kim.

**Validation:** A. Ram Hong.

**Writing – original draft:** Jee Hee Yoon.

**Writing – review & editing:** Jee Hee Yoon, Ho-Cheol Kang.

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
