## [Decision Letter · Decision Letter 0]

15 Jun 2020

PONE-D-20-13614

Unusual metastases from differentiated thyroid cancers: A multicenter study in Korea

PLOS ONE

Dear Dr. Kang,

Thank you for submitting your manuscript to PLOS ONE. After careful consideration, we feel that it has merit but does not fully meet PLOS ONE’s publication criteria as it currently stands. Therefore, we invite you to submit a revised version of the manuscript that addresses the points raised during the review process.

We look forward to receiving your revised manuscript.

Kind regards,

Domenico Albano

Academic Editor

PLOS ONE

Journal Requirements:

2. In your manuscript you have stated that the study was approved by the institutional review boards of the participating hospitals. Please include this information in the ethics statement on the online submission form.

3. In the ethics statement in the manuscript and in the online submission form, please provide additional information about the patient records used in your retrospective study.

Specifically, please ensure that you have discussed whether all data were fully anonymized before you accessed them and/or whether the IRB or ethics committee waived the requirement for informed consent.

If patients provided informed written consent to have data from their medical records used in research, please include this information.

4. Please state the names of the four hospitals included in this study in the methods section of your main manuscript (currently this information is only found in your abstract).

6. One of the noted authors is a group: Korean Thyroid Cancer Study Group.

In addition to naming the author group, please list the individual authors and affiliations within this group in the acknowledgments section of your manuscript.

Please also indicate clearly a lead author for this group along with a contact email address.

Additional Editor Comments:

Dear authors,

I suggest to revise the paper following reviewers suggestions

Reviewers' comments:

Reviewer's Responses to Questions

**Comments to the Author**

1. Is the manuscript technically sound, and do the data support the conclusions?

Reviewer #1: Yes

Reviewer #2: Partly

2. Has the statistical analysis been performed appropriately and rigorously? 

Reviewer #1: Yes

Reviewer #2: Yes

3. Have the authors made all data underlying the findings in their manuscript fully available?

Reviewer #1: Yes

Reviewer #2: Yes

4. Is the manuscript presented in an intelligible fashion and written in standard English?

Reviewer #1: Yes

Reviewer #2: Yes

5. Review Comments to the Author

Reviewer #1: Congratulations for presenting a very interesting paper concerning unusual metastases of differentiated thyroid cancers in Korea. The authors presented adequate methodology. The results were analyzed in details.

I have one notice. Though the authors used many abbreviations in Abstract (DTC,FDG, PET, CT, PTC, TKI, WBS, Tg) not necessarily obvious for all readers, however the use of an unknown shortcut “UMs” is unacceptable. All the more, descriptions for abbreviations should not be included here in Abstract. Explanations of abbreviations cannot be duplicated both in Abstract and Introduction section.

Reviewer #2: The present study highlights an important yet less reported and understood area of differentiated thyroid malignancies. This is in part due to rarity of such findings.

There are however, few area on which the authors can improve on their manuscript.

Methods:

Study design should be explicitly mentioned.

Similarly, study duration can be mentioned to enhance readers understanding on the data gathered.

Results:

Total number of thyroid patients screened can be added.

This can further lead to computation of the rate of UMs in their cohort.

Table 3 needs revision: Mortality was assessed in months/years?

Discussion:

There are a number of case series looking into unusual presentation of differentiated thyroid metastasis, they are basically, synchronous metastasis. Comparison with these studies could be helpful to improve on discussion.

This was a large multi-center study, addition of a comparison arm (usual distant metastasis to lung and bone) such as a case control study could add to the strength of the study.

6. PLOS authors have the option to publish the peer review history of their article (what does this mean?). If published, this will include your full peer review and any attached files.

Reviewer #1: No

Reviewer #2: No

---

## [Author Response · Author response to Decision Letter 0]

4 Aug 2020

Response to Reviewer 1’s comment

Reviewer #1: Congratulations for presenting a very interesting paper concerning unusual metastases of differentiated thyroid cancers in Korea. The authors presented adequate methodology. The results were analyzed in details.

I have one notice. Though the authors used many abbreviations in Abstract (DTC,FDG, PET, CT, PTC, TKI, WBS, Tg) not necessarily obvious for all readers, however the use of an unknown shortcut “UMs” is unacceptable. All the more, descriptions for abbreviations should not be included here in Abstract. Explanations of abbreviations cannot be duplicated both in Abstract and Introduction section.

ANSWER:

Thank for your generous review and good comment. We agree with you that acronyms in the abstract detract from the readability of the paper, so we have deleted all abbreviations in the abstract to improve readability and to avoid duplication.(line 21,23,25,26,28,29,31,32,33,34,35,38,39,40,41,42,43,45,46,50,52).

Response to Reviewer 2’s comment

Reviewer #2: The present study highlights an important yet less reported and understood area of differentiated thyroid malignancies. This is in part due to rarity of such findings.

There are however, few area on which the authors can improve on their manuscript.

1. Methods:

Study design should be explicitly mentioned.

Similarly, study duration can be mentioned to enhance readers understanding on the data gathered.

ANSWER:

Thank for your meticulous review. We added more details of study design including the study duration. We screened medical records of DTC patients who had been treated between January 2000 and August 2016 from four tertiary hospitals in Korea and searched DTC patients with distant metastasis including usual (lung, bone, and brain) and unusual metastasis (UM). (line 78-82: We screened DTC patients who had been treated between January 2000 and August 2016 from four tertiary hospitals in Korea (Chonnam National University Hwasun Hospital, Asan Medical Center, Busan National University Hospital, Severance Hospital) and retrospectively reviewed the medical records of DTC patients with distant metastasis. Distant metastases were divided to two groups; usual metastasis and UM.)

2. Results:

(1) Total number of thyroid patients screened can be added.

 This can further lead to computation of the rate of UMs in their cohort.

ANSWER:

According to your recommendation, we have added the total number of DTC patient screened (n=38,772) and calculated the rate of UMs in our cohort. (line 117-119: Among investigated total of 38,772 DTC patients, 442 usual distant metastases (1.1%) in 392 patients (321 cases of lung metastases, 105 cases of bone metastases, and 16 cases of brain metastases) and 25 UMs (0.06%) in 19 patients were observed.)

Compared to other researcher’s reports, our data on distant metastasis seems to be rarer, so we added sentences to explain why distant metastasis is rarer in our cohort by citing Korea Central Cancer Registry data. (line 231-236: The National Epidemiologic Survey of Thyroid cancer (NEST) study based on the Korea Central Cancer Registry demonstrated 34 distant metastases (0.6%) among total 5,796 thyroid cancer patients during 10.6 years of follow-up (16). So, our data is in accordance with Korean nationwide data. The low incidence of distant metastases might by due to early surgery and high incidence of small papillary microcarcinoma in Korea.)

(2) Table 3 needs revision: Mortality was assessed in months/years?

ANSWER:

We presented the median follow-up duration from UM diagnosis (months)(IQR), however it is not easy to recognize. So, we changed it to Median follow-up months from UM diagnosis (IQR) (Table 3).

3. Discussion:

(1) There are a number of case series looking into unusual presentation of differentiated thyroid metastasis, they are basically, synchronous metastasis. Comparison with these studies could be helpful to improve on discussion.

ANSWER:

As you mentioned, there have been several case reports and series of UM in DTC patients which occurred synchronously. Even though direct comparison of synchronous and metachronous UMs in DTC patients is impossible due to the lack of basic data in previous studies, some of synchronous metastasis have been published as case reports showing the difficulty in finding the primary cancer focus. We added this point in line 237-239: Some case reports and series have shown that synchronous UM in DTC patients presenting initially with UM-related symptoms, but they have difficulty in finding the primary cancer origin due to unusual metastatic sites (17).

In previous studies (Zunino et al and Madani et al), the incidence of metachronous UMs were higher than those of synchronous UMs. This is the same point of view as our study. Long term follow-up due to relatively lower mortality of DTC patients give us more chance to find UMs from DTC. Additionally, high quality of imaging studies used in follow-up period recently increased the probability to discover UMs. Only 4 patients (21.1%) in our study had synchronous UM, indicating a higher incidence of UMs in advanced thyroid cancer during long-term follow-up with diverse imaging modalities such as PET-CT, 131I WBS, and CT scans due to a metachronous pattern.

(2) This was a large multi-center study, addition of a comparison arm (usual distant metastasis to lung and bone) such as a case control study could add to the strength of the study.

ANSWER:

We appreciate your kind and meaningful suggestion; however we are not able to do such a case-control study due to the limitations in the data set.

---

## [Decision Letter · Decision Letter 1]

12 Aug 2020

Unusual metastases from differentiated thyroid cancers: A multicenter study in Korea

PONE-D-20-13614R1

Dear Dr. Kang,

We’re pleased to inform you that your manuscript has been judged scientifically suitable for publication and will be formally accepted for publication once it meets all outstanding technical requirements.

Kind regards,

Domenico Albano

Academic Editor

PLOS ONE

**Comments to the Author**

1. If the authors have adequately addressed your comments raised in a previous round of review and you feel that this manuscript is now acceptable for publication, you may indicate that here to bypass the “Comments to the Author” section, enter your conflict of interest statement in the “Confidential to Editor” section, and submit your "Accept" recommendation.

Reviewer #1: All comments have been addressed

2. Is the manuscript technically sound, and do the data support the conclusions?

Reviewer #1: Yes

3. Has the statistical analysis been performed appropriately and rigorously? 

Reviewer #1: Yes

4. Have the authors made all data underlying the findings in their manuscript fully available?

Reviewer #1: Yes

5. Is the manuscript presented in an intelligible fashion and written in standard English?

Reviewer #1: Yes

6. Review Comments to the Author

Reviewer #1: Thank you for carefully analyzing all comments and making appropriate corrections to the manuscript. Indeed, ther is a clear flow reading your manuscript.

7. PLOS authors have the option to publish the peer review history of their article (what does this mean?). If published, this will include your full peer review and any attached files.

Reviewer #1: No

---

## [Editor Report · Acceptance letter]

14 Aug 2020

PONE-D-20-13614R1 

Unusual metastases from differentiated thyroid cancers: A multicenter study in Korea 

Dear Dr. Kang:

I'm pleased to inform you that your manuscript has been deemed suitable for publication in PLOS ONE. Congratulations! Your manuscript is now with our production department. 

Kind regards, 

on behalf of

Dr. Domenico Albano 

Academic Editor

PLOS ONE